# Unlocking Casein Bioactivity: Lactic Acid Bacteria and Molecular Strategies for Peptide Release

**DOI:** 10.3390/ijms26178119

**Published:** 2025-08-22

**Authors:** Chenxi Huang, Lianghui Cheng

**Affiliations:** 1Key Laboratory of Combinatorial Biosynthesis and Drug Discovery (Ministry of Education), School of Pharmaceutical Sciences, Wuhan University, Wuhan 430071, China; chenxihuang@whu.edu.cn; 2Institute of Intelligent Sport and Proactive Health, Department of Health and Physical Education, Jianghan University, Wuhan 430056, China

**Keywords:** β-casein, bioactive peptides, lactic acid bacteria, molecular tools, CRISPR-Cas, peptide transporters, functional food

## Abstract

Bioactive peptides encrypted in bovine β-casein display diverse physiological functions, including antihypertensive, antioxidative, antimicrobial, and immunomodulatory activities. These peptides are normally released during gastrointestinal digestion or microbial fermentation, especially by proteolytic systems of lactic acid bacteria (LAB). However, peptide yields vary widely among LAB strains, reflecting strain-specific protease repertoires. To overcome these limitations, the scientific goal of this study is to provide a comprehensive synthesis of how synthetic biology, molecular biotechnology, and systems-level approaches can be leveraged to enhance the targeted discovery and production of β-casein-derived bioactive peptides. Genome engineering tools such as clustered regularly interspaced short palindromic repeats associated system (CRISPR/Cas) systems have been applied to modulate gene expression and metabolic flux in LAB, while inducible expression platforms allow on-demand peptide production. Additionally, cell-free systems based on LAB lysates further provide rapid prototyping for high-throughput screening. Finally, multi-omics approaches, including genomics, transcriptomics, proteomics, and metabolomics, further help pinpoint regulatory bottlenecks and facilitate rational strain optimization. This review provides a comprehensive overview of bioactive peptides derived from bovine β-casein and highlights recent progress in LAB-based strategies—both natural and engineered—for their efficient release. These advances pave the way for developing next-generation functional fermented foods enriched with targeted bioactivities.

## 1. Introduction

Fermented dairy products are produced through the fermentation of milk. Buttermilk, cream, butter, yogurt, and cheese are among the most well-known examples [1]. Milk remains one of the most widely consumed foods globally, as reported in OECD–FAO Agricultural Outlook 2024–2033, world milk production—around 81% from cows, 15% from buffaloes, and 4% from goats, sheep, and camels—is forecast to grow at 1.6% per year over the next decade, reaching 1085 million tons by 2033 [2]. Dairy products exhibit remarkable diversity due to the complex composition of milk and the wide range of microorganisms that can thrive in it. Dairy products are a valuable source of essential nutrients, including high-quality proteins, omega-3 fatty acids, oleic acid, conjugated linoleic acid, vitamins (e.g., A, D, B12), minerals (e.g., calcium, phosphorus, zinc), and bioactive peptides [3]. In addition to their nutritional benefits, products such as yogurt and cheese offer appealing organoleptic qualities—namely, desirable texture and flavor—that enhance the human daily diet [4] (Figure 1a).

The proteins in milk are broadly classified into two main groups, caseins and whey proteins. Approximately 80% of the total protein content in bovine milk is composed of caseins, with the rest being whey proteins. According to their primary amino acid sequence homology, bovine caseins can be divided into four subtypes: αs1-casein, αs2-casein, β-casein, and κ-casein (Figure 1b). The whey protein fraction mainly consists of α-lactalbumin, β-lactoglobulin, lactoferrin, immunoglobulins, serum albumin, and the secretory component [5]. Casein can be utilized in its native form or processed into caseinates and peptides. In particular, hydrolysates of β-Casein have drawn increasing attention for their potential applications in functional foods, owing to the bioactive peptides they release [6]. Typically, these peptides consist of 2 to 20 amino acids (AAs) [7]. Within the precursor protein sequence, they are inactive until liberated by (i) digestion in the gastrointestinal tract [7], (ii) microbial fermentation involving proteolytic microorganisms [8], or (iii) enzymatic hydrolysis using plants or microorganisms [9]. Once released, these peptides may exert various beneficial effects on the cardiovascular, immune, nervous, and/or human digestive system [10] (Figure 1c).

Among the four casein subtypes, β-casein stands out as a particularly effective precursor for generating bioactive peptides due to several favorable structural characteristics. Its primary sequence is particularly rich in proline, glutamine, and phosphorylated serine residues, which are commonly found in bioactive motifs associated with antioxidant, antihypertensive, immunomodulatory, or mineral-binding properties [11]. Moreover, β-casein exhibits a relatively open and flexible tertiary structure due to its high proportion of hydrophilic residues, making it more accessible to proteolytic cleavage compared to other milk proteins. These features facilitate the targeted release of functional peptides, both in vivo and under controlled fermentation processes [12,13].

The enzymatic accessibility of β-casein makes it an ideal substrate for controlled proteolysis during dairy fermentation. In this context, lactic acid bacteria (LAB) play a central role in the proteolytic breakdown of caseins during dairy fermentation. LAB possess a sophisticated proteolytic system composed of cell-envelope proteinases, intracellular peptidases, and peptide transporters, which allow LAB to hydrolyze caseins into small peptides and free amino acids [14,15]. This natural capacity not only contributes to the development of flavor and texture in fermented dairy products but also provides a biological route for releasing health-promoting peptides from milk proteins. However, spontaneous peptide production by wild-type LAB is often inefficient, strain-dependent, and rarely targeted toward sequences with specific bioactivities.

To overcome these limitations, a suite of molecular strategies—including genetic engineering, synthetic biology, and omics-guided strain optimization—has been increasingly employed to enhance the proteolytic capacity and peptide specificity of LAB [16]. These strategies enable the precise modification of key enzymes, transport systems, and regulatory elements involved in casein hydrolysis. As a result, engineered LAB strains are now emerging as powerful biocatalysts for the tailored production of bioactive peptides from β-casein in a controlled, food-grade manner. The integration of such molecular tools into LAB-based systems represents a promising avenue for developing next-generation functional dairy ingredients and fermented nutraceuticals.

**Figure 1 ijms-26-08119-f001:**
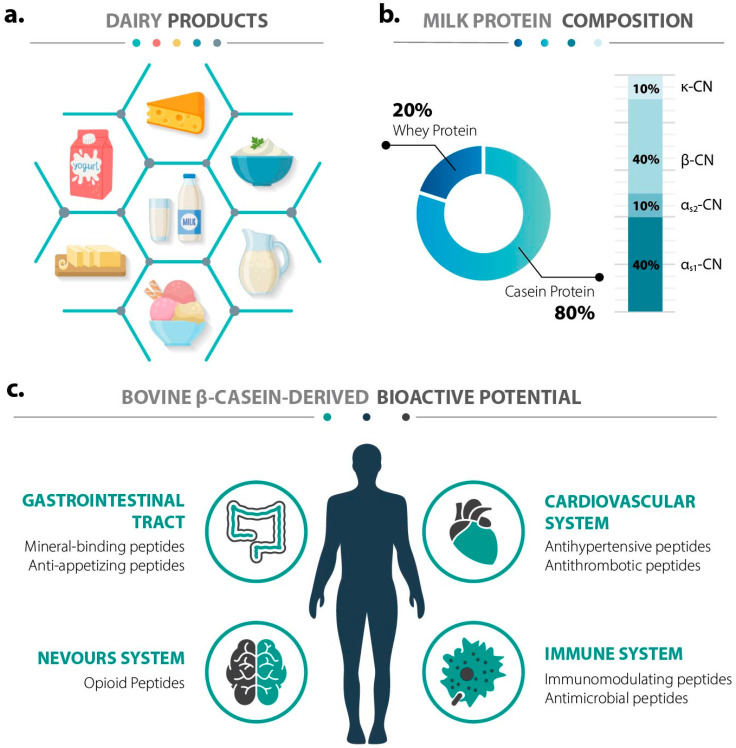
(**a**) Representative milk-derived (dairy) products. Clockwise, starting at noon: cheese, sour cream, soured or buttermilk, ice cream, butter, yogurt. In the center: the raw starting material, milk. (**b**) Pie chart showing the milk protein composition and percentage ratios; the stacked bar chart displays the casein composition and percentage distribution [17]. (**c**) Bovine β-casein-derived peptide bioactivities benefiting human health.

## 2. Functional Diversity of Bioactive Peptides from Bovine β-Casein

Bovine β-casein is a rich source of encrypted peptides with diverse biological activities, which are released during digestion or microbial fermentation. These peptides can target major physiological systems, including the gastrointestinal, cardiovascular, nervous, and immune systems (Figure 1c) [6,11]. The functional diversity of β-casein-derived peptides reflects differences in their amino acid composition, structure, and susceptibility to proteolytic cleavage. This section summarizes the major classes of bioactive peptides derived from bovine β-casein and their corresponding physiological roles.

### 2.1. Caseinophosphopeptides (CPPs): Mineral-Binding and Immunomodulatory Peptides

Among the different functional peptides, caseinophosphopeptides (CPPs) are well-characterized for their mineral-binding capabilities. Derived from casein, CPPs are phosphorylated peptides capable of binding and solubilizing minerals, including zinc and calcium [18]. Phosphate groups are attached to serine residues through monoester bonds, with bovine β-casein containing five phosphorylatable serines. Two major CPPs identified from β-casein are fragment 16–40 (f(16–40)) and fragment 16–43 (f(16–43)) [19,20,21,22]. In rats ingesting β-casein, the peptide f(16–40) has been detected in the small intestine, where it elevates soluble calcium concentrations in the distal ileum and facilitates passive calcium absorption [20]. In addition, f(16–40) promotes the proliferation of mouse spleen cells and rabbit Peyer’s patch cells [22]. Similarly, the peptide f(16–43) demonstrates immune-stimulatory properties, including stimulation of immunoglobulin production and cytokine secretion in human immune cells, as well as increased proliferation and interleukin-6 (IL-6) expression in B lymphocytes [19,20].

### 2.2. β-Casomorphins (β-CMs): Opioid-like Peptides

Another major category of β-casein-derived peptides is β-casomorphins (β-CMs), which possess opioid-like activity. These peptides act primarily as μ-opioid receptor agonists and have been linked to various biological functions [23,24]. The heptapeptide β-CM7, the first exogenous opioid peptide discovered [25], not only exhibits opioid activity but also enhances mucin production in the gastrointestinal tract through the µ-opioid pathway [26]. It also shows hypoglycemic effects in diabetic rats by increasing plasma insulin levels and shows increased angiotensin-converting enzyme (ACE) inhibitory activity [27,28]. The release of β-CM7 varies among milk types, with A1 milk generating more than A2, which may explain differential effects on gut and neurological health [29]. Recent studies have further highlighted the role of β-CM7 in gut–brain axis modulation by affecting gut permeability, inflammation, and central nervous system signaling [30]. Another peptide, β-CM5, is noted as the strongest opioid agonist among β-CMs and exhibits high resistance to proteolytic digestion. It has been shown to alter gut motility in rats [31,32] and stimulate neurite outgrowth in neuronal cell lines [33], as well as improve learning and memory disturbances related to cholinergic dysfunction [34].

### 2.3. DPP-IV Inhibitory Peptides: Metabolic Regulation

In addition to opioid activity, certain peptides derived from β-casein have shown potential in metabolic regulation [35]. Of particular interest are peptides that inhibit dipeptidyl peptidase-4 (DPP-IV), an enzyme that degrades incretin hormones like glucose-dependent insulinotropic polypeptide (GIP) and glucagon-like peptide-1 (GLP-1), which are essential for insulin secretion [36]. To date, 21 DPP-IV inhibitory peptides have been identified from bovine β-casein, ranging from 3 to 9 amino acids with IC_50_ values between 46 and 1300 µM. About half of these peptides originate from fragment f(75–95), and many possess a characteristic proline at the second position from the N-terminus, suggesting a potential structural motif underlying DPP-IV-inhibitory activity. Tripeptides conforming to the X-Pro-X motif function as competitive inhibitors of DPP-IV, for instance, LPL and LPQ display IC_50_ values of 241.4 µM and 82 µM, respectively. Nonetheless, detecting such small peptides using mass spectrometry (MS) remains challenging; advances in Liquid Chromatography–Mass Spectrometry (LC-MS) techniques are facilitating the discovery of additional short DPP-IV inhibitors.

### 2.4. Antimicrobial Peptides (AMPs)

Beyond metabolic regulation, β-casein-derived peptides also exhibit antimicrobial activity [37]. Antimicrobial peptides (AMPs) are generally short, positively charged peptides capable of disrupting microbial membranes. Such peptides from food sources are of interest due to their safety and natural origin [38]. Several antimicrobial peptides have been identified from bovine caseins following enzymatic hydrolysis. For example, two peptides, casecidin-15 and casecidin-17, derived from the C-terminal region of β-casein, showed inhibitory effects against Escherichia coli and exhibited amphiphilic structures that support their antimicrobial function [39]. Furthermore, plasmin digestion of β-casein revealed eight antibacterial peptides, notably βC8(VKEAMAPK), βC12(EAMAPKHK), and βC14(VLPVPQKAVPYPQR), which were highly active against Gram-positive bacteria and characterized by their hydrophobicity and C-terminal basic residues [40]. These findings highlight β-casein as a promising source of natural antimicrobial agents, with potential applications in enhancing the safety and functionality of dairy-based products.

### 2.5. Immunomodulatory Peptides

Several β-casein peptides function as immunomodulators, modulating immune responses without the side effects of synthetic drugs like cyclosporine and glucocorticoids [41], which provides a promising strategy for dietary modulation of the immune response [42,43]. These peptides can enhance lymphocyte proliferation, regulate cytokine expression, and attenuate inflammatory responses. For instance, β-casein-derived peptides such as QEPVL have been demonstrated to alleviate inflammation and restore gut microbiota balance in DSS-induced colitis models, likely through the inhibition of the NLRP3/NF-κB signaling pathway [44].

### 2.6. ACE-Inhibitory Peptides: Antihypertensive Potential

Among the most studied functionalities of β-casein-derived peptides is their ability to inhibit ACE, a key enzyme in blood pressure regulation [45,46,47]. Therefore, ACE inhibitors are helpful agents in the therapeutic treatment of hypertension. Synthetic ACE inhibitors such as captopril and enalapril are effective but often have adverse side effects, prompting interest in natural alternatives [48]. ACE-inhibitory activity is exhibited by more than half of the bioactive peptides derived from bovine β-casein (Figure 2a). About 56% of these peptides display IC_50_ values below 100 μM, whereas fewer than 10% have IC_50_ values in the range from 500 μM to 1000 μM (Figure 2b–d). Notably, the most studied ACE-I tripeptides, VPP and IPP, were identified through the fermentation of skim milk [49] and have been commercialized in Calpis sour milk (Calpis Co., Sumida City, Japan) and Evolus^®^ (Valio, Finland) [50]. The IC_50_ values of VPP and IPP are reported to be 9 μM and 5 μM, respectively, in an in vitro ACE inhibition assay. Identified as the most potent peptide to date, Met-Ala-Pro (MAP) exhibits an IC_50_ of 0.8 μM in an in vitro ACE inhibition assay and has also demonstrated pronounced blood pressure-lowering effects in an in vivo hypertensive rat model (3 mg/kg dose, *p* < 0.01 at 8 h post-administration) [51].

### 2.7. Other Bioactivities

Beyond the aforementioned functions, peptides derived from bovine β-casein also exhibit diverse biological activities, including antioxidant effects [52], prolyl endopeptidase inhibition [53], and blood coagulation inhibition [54]. Recent work further highlights their value in lipid and glucose homeostasis. Jiang et al. [55] isolated β-casein peptides VLPVPQ and LQPE, which lowered micellar cholesterol solubility and suppressed the cholesterol transporter NPC1L1 in Caco-2 cells while upregulating efflux transporters ABCA1 and ABCG8. Similarly, Gong et al. [56] reported that the goat-milk peptide QEPVLGPVRGPFP from β-casein attenuated high-glucose-induced insulin resistance in HepG2 cells by enhancing glucose uptake and glycogen synthesis, while downregulating gluconeogenic genes. Synthetic peptides such as SQSKVLPVPQK also promoted trans-intestinal cholesterol excretion via ABCG5 activation and downregulated hepatic bile acid synthesis through the FGF15/19–LXRα axis [57]. Collectively, these findings underline the hypolipidemic and insulin-sensitizing potential of β-casein–derived peptides, supporting their inclusion in functional foods targeting metabolic disorders.

To catalogue these multifunctional peptides in a consistent and traceable manner, we compiled bovine β-casein-derived bioactive peptides from three widely used databases of milk-derived peptides: BIOPEP [58], MBPDB [59], and EROP-Moscow [60], retrieving a total of 176 entries. To ensure data quality and relevance, we applied four exclusion criteria: (i) no supporting reference; (ii) only hypothetical or predicted activity; (iii) discrepancies between database sequences and original literature; and (iv) bitter taste reported without any functional bioactivity. For instance, peptides such as PGP, PHQ, FPPQS, VLP, and GPV were present only in BIOPEP without literature support, while MBPDB recorded LVYPFPGPIP as an ACE inhibitor, although the cited publication reported activity for LVYPFPGPIH. After applying these quality control steps, 140 unique β-casein-derived bioactive peptides were retained and are listed in Table 1. A comprehensive diagram summarizing the theoretical curation process and exclusion steps is presented in Figure 3.

## 3. Proteolytic Systems of Lactic Acid Bacteria and Natural Peptide Liberation from β-Casein

Having established the diverse bioactivities of β-casein–derived peptides and their promising roles in metabolic and cardiovascular health, it is essential to understand how these peptides are generated at the molecular level. In dairy fermentations, lactic acid bacteria (LAB) are the primary microbial agents responsible for protein breakdown and peptide release. These Gram-positive, low GC-content bacteria are widely used in the fermentation of milk, meat, fish, and vegetables, and are classified as “generally recognized as safe (GRAS)” by the U.S. Food and Drug Administration (FDA) [14,164]. Since their first industrial application in cheese and sour milk in the 1890s, LAB have gained increasing economic and scientific interest [165,166,167]. Beyond acidification and flavor development, LAB have also been used as biocatalysts for the release of bioactive peptides, such as antihypertensive peptides, antimicrobial peptides, and immunomodulatory peptides [168]. One of the most relevant features of LAB in dairy fermentation is their ability to release bioactive peptides from β-casein via a sophisticated proteolytic system. Their proteolytic machinery not only serves nutritional needs for bacterial growth but also drives the liberation of bioactive sequences embedded in food proteins. This section outlines the molecular mechanisms involved and highlights the diversity of peptides generated.

### 3.1. Extracellular Proteolysis of β-Casein

Casein is degraded by dairy LAB through a proteolytic system to release amino acids necessary for cell growth (Figure 4). Simultaneously, the freed amino acids and small peptides play a role in the sensory characteristics of the dairy product. Among LAB, *Lactococcus lactis* is the most studied model for proteolysis. Its proteolytic system initiates casein degradation through a cell-envelope proteinase, which hydrolyses the extracellular protein (β-casein in this case) into the oligopeptides of varying length [14,15]. Specifically, the PrtP enzyme, which encodes the cell-envelope proteinase PrtP, is preceded by the divergently transcribed *prtM* gene, which encodes a lipoprotein, PrtM, which is anchored in the cell membrane and is required for the maturation and activation of PrtP [169]. Interestingly, while both *prtP* and *prtM* genes are present in *Lacticaseibacillus paracasei* [170], only *prtP* is found in *Lactobacillus helveticus*, *Lb delbrueckii* subsp. *bulgaricus*, and *Lc. thermophilus* [171,172], indicating species-specific differences in the proteolytic system.

### 3.2. Peptide Transport and Intracellular Degradation

Following initial cleavage of extracellular β-casein into oligopeptides, transport systems such as Opp, DtpT, and Dpp mediate peptide uptake: Opp handles peptides of 5–35 amino acids, whereas DtpT and Dpp are specialized for di- and tripeptides [173]. Once internalized, peptides will be further degraded in the cytoplasm by a suite of 15 intracellular peptidases with defined cleavage specificities [15]. Residual amino acids and peptides not utilized by the cell are released into the environment, where some of the peptides have potent biological functions [17].

### 3.3. Release of Functional Peptides

Certain LAB strains have demonstrated the ability to release bioactive peptides from food proteins, especially from caseins. The extent of hydrolysis and the resulting bioactivity are influenced both by the proteolytic enzymes expressed by the strains and the amino acid sequences of various casein types. As an example, an antimicrobial peptide released from human β-casein by *Lb. helveticus* PR4 exhibits activity against a broad spectrum of pathogens, such as *Enterococcus faecium*, *Salmonella* spp., *St. aureus*, *E. coli*, *Bacillus megaterium*, *Yersinia enterocolitica*, and *L. innocua* [78]. Moreover, milk fermented with *Bifidobacterium bifidum* MF 20/5, which is isolated from the Japanese probiotic product Bion-3^®^, was found to contain the peptide VLPVPQK with antioxidant activity [82]. In addition, LC analysis of Gouda-style cheese identified four antioxidative peptides characterized by an N-terminal X-Pro motif, all displaying IC_50_ values below 200 μM. Among these, LPQNIPPL exhibited the strongest antioxidative activity [104].

Beyond antimicrobial and antioxidant activities, LAB can also liberate immunomodulatory peptides. For example, milk fermented by *Lb. helveticus* LH-2 was shown to stimulate murine macrophage activity; four novel immunomodulatory peptides were subsequently identified [174]. Other strains, including *Enterococcus faecalis* CECT5727 and BCS27, *Lb. delbrueckii*, *Lb. helveticus*, *Sa. cerevisiae*, *Lb. delbrueckii* subsp. *bulgaricus* SS1, *Lb. delbrueckii* subsp. *lactis*, and *Lc. lactis* subsp. *cremoris* FT4 can generate ACE-inhibitory peptides from casein, demonstrating the role of LAB as biocatalysts for natural antihypertensive compounds (Table 1) [50,69].

Taken together, these findings illustrate the complexity and efficiency of LAB proteolytic systems in releasing functional peptides from β-casein. Understanding these microbial mechanisms provides a foundation for selecting or engineering strains optimized for peptide production in functional dairy and nutraceutical applications. However, when redirected toward the intentional production of specific bioactive peptides, native LAB systems reveal notable limitations. These include strain-dependent variability in protease activity, low or inconsistent yields of target peptides, and limited specificity for cleaving desired sequences. In addition, uncontrolled proteolysis can result in over-hydrolysis, degrading functional peptides and reducing their bioactivity. This unpredictability hampers both mechanistic studies and industrial applications, where reproducibility, efficacy, and scalability are essential. These challenges underscore the need for rationally designed LAB strains with greater control over proteolytic precision. In the following section, we highlight emerging molecular strategies—such as genome editing, synthetic biology, and omics-driven optimization—designed to overcome these limitations and fully harness LAB’s biosynthetic potential for targeted bioactive peptide production.

## 4. Molecular Strategies for Enhancing LAB-Mediated Liberation of Bioactive Peptides

The preceding section illustrated how wild-type LAB release bioactive fragments from β-casein via a sequential route of extracellular proteolysis, peptide transport, and intracellular trimming. However, the native process often under- or over-hydrolyses target peptides. The challenge is, therefore, not only to liberate the desired peptide but also to protect it from further degradation. This section summarizes the molecular toolkit now available to rebalance LAB proteolysis at three levels: promoter-driven control of key enzymes, chromosome-level editing, and high-throughput discovery platforms (Figure 5).

### 4.1. Inducible Systems for Targeted Regulation of Proteolysis

Because *Lc. lactis* is the best-characterized LAB; its proteolytic genes provide a model blueprint for targeted manipulation. Over recent decades, a range of genome engineering technologies for lactic acid bacteria (LAB) has emerged [175]. In *Lc. lactis*, gene expression can be precisely controlled by employing the Nisin-Controlled Expression (NICE) platform, which is widely regarded as the leading inducible expression system for this species. This platform functions through a quorum-sensing two-component regulatory circuit consisting of NisR and NisK proteins. Activation occurs upon the detection of nisin, an antimicrobial peptide naturally synthesized by specific *Lc. lactis* strains [176]. This system offers tight control over gene expression and is widely applied in *Lc. lactis* as a food-grade expression platform, particularly for the production of nisin, an important antimicrobial peptide used in the food industry.

In addition to the NICE system, alternative inducible platforms have been introduced to diversify regulatory options and avoid potential cross-reactivity. For instance, Zinc has also been employed as an inducer in LAB expression systems. Mu et al. [177] developed a zinc-responsive system named Zirex, which includes the *pneumococcal* repressor SczA and the P_czcD_ promoter. This system enables gene expression under elevated Zn^2+^ concentrations. Another notable system is the xylose-inducible expression system (XIES), in which the P_xylT_ promoter is activated by xylose but repressed by more common sugars such as glucose, fructose, and mannose. This sugar-switchable system is particularly attractive in industrial applications due to the low cost and non-toxic nature of xylose as an inducer [178]. More recently, an agmatine-inducible expression system has been introduced, composed of the agmatine-sensing transcriptional activator AguR and the PaguB promoter. This system offers tight control and dose-dependent gene expression, making it a promising complement or alternative to the traditional NICE system [179].

Together, these inducible promoter systems enable precise temporal and quantitative control over genes encoding proteases, transporters, or peptidases, allowing researchers to upregulate or downregulate specific steps in the proteolytic cascade to favor the accumulation of targeted bioactive peptides from β-casein.

### 4.2. Genome Editing Tools for Stable Manipulation of Proteolytic Pathways

Beyond transient expression control, stable genome editing provides a powerful avenue to reprogram LAB strains for optimized peptide release. Gene knockout can be accomplished in *Lc. lactis* by the double crossover method using the non-replicative pORI integration vector [180] or plasmid pCS1966 [181]. Although precise, these traditional methods are time-consuming and labor-intensive, often requiring several weeks to obtain a single marker-free mutant.

To accelerate strain development, novel genome engineering technologies have also been established in LAB. These include bacteriophage-derived recombinase enzymes for single-stranded DNA recombineering in *Lb. reuteri* and *Lc. lactis* [182], which facilitates single-stranded DNA recombineering and enables site-specific nucleotide substitutions with higher efficiency. The clustered regularly interspaced short palindromic repeats associated system (CRISPR/Cas) genome editing technology, which combines clustered regularly interspaced short palindromic repeats with Cas proteins, has emerged as one of the most powerful and widely adopted gene editing tools, particularly for eukaryotic organisms where alternative targeted approaches are limited [183]. Successful applications of the CRISPR/Cas9 system have also been reported in various bacterial genera, including *Escherichia*, *Bacillus*, *Streptococcus*, and *Clostridium* [184,185,186,187]. Within LAB, a CRISPR/Cas9-assisted single-stranded DNA recombineering system in *Lb. reuteri* employed Cas9-induced double-strand breaks (DSBs) as a counter-selection mechanism [188]. Furthermore, an inducible dual-promoter plasmid was constructed in Lc lactis, building on the previously described NICE system. This plasmid harbors the *dCas9* and *sgRNA* genes under the control of the nisin-inducible promoter PnisA and was successfully utilized to repress transcription of the target gene *htrA* [189].

However, conventional genome editing in *Lc. lactis* typically relies on the RecA-mediated double-crossover recombination using non-replicative or conditionally replicative plasmids [181,190]. Gene knockout or insertion in *Lc. lactis* typically follows a two-step protocol—vector integration followed by co-integrate resolution—which generally requires approximately three weeks to generate a single mutant. The resulting marker-free strain can then serve as the starting point for subsequent rounds, allowing the introduction of multiple mutations into the same genetic background. However, this approach is limited by the considerable time needed to obtain a strain carrying multiple mutations. Leveraging the powerful CRISPR/Cas-based tools to engineer the proteolytic system of Lc. lactis could shorten the time required and lessen the workload. Moreover, utilizing CRISPR/dCas9, rather than CRISPR/Cas9, is more appropriate for Lc. lactis, as this organism lacks a non-homologous end joining (NHEJ) pathway and therefore cannot repair double-strand DNA breaks.

To overcome this, researchers have developed several alternative strategies. One widely adopted approach employs the *Streptococcus pyogenes* Cas9^D10A^ nickase (SpCas9^D10A^), which introduces single-strand breaks (SSBs) instead of DSBs, thereby reducing cytotoxicity. In *Lb. casei* LC2W, a system combining SpCas9^D10A^ with target-specific sgRNA and repair templates successfully circumvented DSB lethality [191]. Similarly, a single-plasmid SpCas9^D10A^-based editing system was developed for the probiotic *Companilactobacillus crustorum*, incorporating an SpCas9^D10A^ expression cassette, the repD/repE replicon, ColE1 origin, sgRNA scaffold, and antibiotic resistance markers [192]. A comparable single-plasmid platform was also established in *Lb. acidophilus* NCFM, featuring a Cas9^D10A^ gene under a constitutive promoter (P6), along with sgRNA and donor templates, enabling gene deletion and insertion [193]. These findings suggest that the exogenous CRISPR-Cas9^D10A^ system can function in LAB even when native CRISPR-Cas machinery is present, highlighting its potential portability to other health-promoting and industrially relevant *Lactobacillus* species.

### 4.3. Cell-Free Protein Synthesis for Rapid Discovery and Screening

While gene regulation and genome editing empower rational strain design, cell-free systems offer complementary platforms for discovering and applying bioactive peptides from LAB. For example, cell-free culture supernatants (CFCS) of LAB, which contain bacteriocins and other metabolites, have been shown to inhibit Listeria monocytogenes in milk, meat, and broth via class IIa bacteriocins like leucocin, pediocin, and sakacin, though effectiveness varies with food matrix and storage conditions [194]. Similarly, the treatment of ribbonfish fillets with LAB-CFCS improved physicochemical quality, extended shelf life, and reduced spoilage at both refrigeration and room temperature (Das et al., 2021) [195]. LAB-CFCS also inhibited Staphylococcus aureus growth and biofilm formation by disrupting adhesion, matrix production, and metabolic pathways [196].

Beyond harvesting natural products, cell-free protein synthesis (CFPS) systems allow for in vitro engineering and screening of synthetic pathways. For instance, a CFPS platform was developed based on the nisin biosynthetic pathway to overcome the toxicity and complexity associated with producing lanthipeptides in living cells. Using this platform, researchers could mine the NCBI database for nisin analogs and, within a single day, identify four novel lanthipeptides with antibacterial activity, demonstrating its high-throughput capacity. Moreover, the CFPS was coupled with a functional assay against gram-negative bacteria, leading to the discovery of a potent nisin mutant (M5) with enhanced activity. Importantly, the CFPS-guided approach not only accelerated discovery but also improved the in vivo titers of nisin and its analogs by informing strain engineering. Given the conserved biosynthetic principles of lanthipeptides, this platform provides a broadly applicable and versatile method for discovering and overproducing a wide range of bioactive peptides beyond nisin [197]. These complementary strategies demonstrate the power of cell-free approaches to discover, optimize, and apply LAB-derived bioactive peptides, especially those challenging to express in living hosts.

### 4.4. Multi-Omics Strategies for System-Level Optimization

In addition to targeted molecular engineering, multi-omics technologies offer a powerful systems-level approach to further optimize the release of bioactive peptides from β-casein by LAB. While traditional methods often rely on single-strain cultures and gene-targeted analysis, multi-omics enables the resolution of dynamic molecular interactions that underlie peptide formation from bovine β-casein in complex environments [198]. Multi-omics technologies—spanning genomics, transcriptomics, proteomics, and metabolomics—provide a powerful systems-level approach to understanding and optimizing the release of bioactive peptides by LAB.

Recent multi-omics studies have demonstrated that mixed LAB fermentation (e.g., *Lactobacillus plantarum* and *L. acidophilus*) can break down structurally complex substrates like bee pollen and activate amino acid metabolic pathways [199]. In broader fermented food systems, multi-omics has been used to map microbial contributions to metabolite production and to reveal strain-specific proteolytic capacities that influence functional peptide profiles [200]. Moreover, in broader fermented food systems, multi-omics has been utilized to map microbial contributions to metabolite production and to reveal strain-specific proteolytic capacities that influence functional peptide profiles. For instance, metabolomic and peptidomic analyses have been used to identify bioactive peptides released during fermentation with novel probiotics like *Lacticaseibacillus rhamnosus* and *Lactiplantibacillus plantarum*, which revealed significant upregulation of ACE-I peptides. In a study of fresh cheese fermented with three novel probiotics, 112 bioactive peptides were significantly upregulated, highlighting the potential of metabolomics and peptidomics for understanding peptide biosynthesis [201]. Additionally, an integrated approach using metabolomics and peptidomics was employed to delineate the characteristic metabolites and peptides in milk fermented with *Lpb. plantarum* L3. This study revealed that *Lpb. plantarum* L3 secreted a complex array of peptidases that cleaved bovine caseins at specific sites to release bioactive peptides, including ACE-I peptides, antioxidant peptides, and antimicrobial peptides. The enzyme systems secreted by *Lpb. plantarum* L3 preferentially cleaved lysine-based peptide bonds, and the released peptides exhibited various bioactivities [202].

These findings underscore the importance of multi-omics approaches in optimizing fermentation conditions to enhance the functional properties of fermented dairy products. By combining the metabolomic and peptidomic data, this integrated multi-omics approach provides valuable insights into the complex enzymatic processes that regulate the release of bioactive peptides during fermentation. Such knowledge can be applied to optimize LAB strains for specific peptide production, leading to improved functional foods with targeted health benefits.

The convergence of inducible promoters, CRISPR toolkits, CFPS prototyping, and multi-omics analytics now enables precision fermentation: LAB strains engineered to release specific β-casein peptides at an industrial scale while minimizing unwanted side-products. These molecular strategies not only enhance functional-food development but also expand LAB’s potential as GRAS biotherapeutic platforms.

## 5. Challenges, Regulatory Considerations, and Future Perspectives

While recent advances in molecular engineering, high-throughput screening, and multi-omics have significantly expanded our ability to discover and produce bioactive peptides from bovine β-casein using LAB, the successful industrial translation of these peptides into food-grade ingredients or supplements remains limited. One of the primary technical challenges lies in peptide stability [203]. Bioactive peptides are often highly susceptible to degradation by gastrointestinal digestion or during food processing procedures such as heating, pH fluctuation, or enzymatic cross-reactions with other food matrix components. This instability may significantly compromise their bioavailability and health-promoting effects in vivo.

Moreover, large-scale production of bioactive peptides using LAB fermentation involves multifaceted optimization. These include ensuring strain stability, maintaining robust proteolytic activity across batches, and avoiding the accumulation of bitter peptides or off-flavors during extended fermentation [204]. The dynamic nature of microbial proteolysis, driven by environmental factors such as substrate concentration, fermentation pH, oxygen levels, and nutrient availability, further complicates standardized production [172]. Strain-specific differences in proteolytic specificity further complicate process standardization and require careful strain selection or genetic tailoring. Moreover, the downstream steps—such as peptide concentration, purification, and formulation for delivery—are also resource-intensive and limit scalability. The use of genetically modified lactic acid bacteria (GM-LAB) offers a promising approach to addressing many of these issues. Through synthetic biology tools such as CRISPR/Cas9 or integrative plasmid systems, LAB can be engineered to enhance their proteolytic systems, selectively release target peptides, or eliminate bitter peptide byproducts. Furthermore, GM-LAB can be designed to selectively produce desired bioactive peptides, facilitating easier downstream purification.

However, the use of genetically modified LAB (GM-LAB) introduces regulatory and consumer perception challenges. Regulatory frameworks vary significantly across regions: in the United States, GRAS designation provides a relatively flexible route for GM strain approval, contingent on evidence of safety. In contrast, the European Union operates the more stringent Qualified Presumption of Safety (QPS), which requires detailed evaluations of each genetically modified strain, including potential toxicity, genetic stability, and absence of transferable antibiotic resistance markers. Additionally, public skepticism toward genetically modified organisms (GMOs), especially in the food and health domains, adds a further layer of complexity. Hence, even for strains with robust safety data, product developers may prefer non-GMO LAB strains to avoid labeling controversies and market resistance, especially in export-oriented and infant products.

Looking forward, the convergence of synthetic biology, artificial intelligence (AI), and precision fermentation is expected to define the next frontier in LAB-based bioactive peptide development. Synthetic biology offers the possibility of designing “next-generation” LAB chassis tailored for controlled proteolysis and defined peptide profiles. To fully realize this potential, interdisciplinary collaboration between food engineers, computational biologists, and fermentation experts will be crucial. Food engineers can bring expertise in optimizing fermentation conditions, scaling up production, and ensuring the consistency of peptide output, while computational biologists will focus on creating predictive models, integrating omics data, and using AI tools to optimize peptide sequence design and bioactivity predictions. This collaboration will ensure that both the fermentation processes and the bioinformatics approaches are aligned to maximize the efficiency and specificity of peptide production. Coupled with multi-omics and high-throughput screening platforms, the rational design of LAB protease repertoires could achieve both diversity and specificity in peptide output. Moreover, food engineers will play a pivotal role in translating computational predictions into practical applications by optimizing fermentation conditions such as nutrient availability, temperature, pH, and oxygen levels, which are crucial for the desired expression of bioactive peptides. Meanwhile, computational biologists, aided by machine learning algorithms, can continually improve their models by analyzing the resulting data from fermentation trials. This dynamic feedback loop will allow for a more iterative and fine-tuned approach to peptide production. AI-driven peptide discovery tools—trained on large datasets of sequence–function relationships—are being increasingly applied to predict peptide bioactivities and optimize sequences with enhanced stability, bioavailability, or receptor binding affinity. The synergy between computational biology and fermentation science will allow for the development of peptides with more predictable and controllable characteristics, ultimately reducing the time and resources required for optimization.

Furthermore, precision fermentation platforms promise reproducible, scalable, and cost-effective peptide production with minimal batch variability. To maximize the impact of these platforms, both computational biologists and food engineers will need to work together to fine-tune the optimization parameters and integrate real-time monitoring technologies, ensuring that the fermentation process remains robust and adaptable to various production scales. In the era of personalized nutrition and microbiome-aware interventions, collaborative efforts between these interdisciplinary teams will also facilitate the development of modular LAB consortia that can be designed to generate peptide profiles specifically tailored to individual health needs or microbiota compositions. This flexibility will allow for personalized approaches to peptide delivery, enhancing the efficacy of bioactive peptides in promoting human health and well-being.

However, despite its scientific advantages, the current implementation of synthetic biology-assisted peptide production using LAB may involve higher upfront costs compared to conventional strain isolation and screening approaches. These expenses primarily arise from two domains: the development of engineered strains, including genetic modification and fermentation process optimization, and the subsequent investment in regulatory compliance, which is essential for product approval and market access. These stages require significant resource input and specialized expertise, posing a barrier to small- and medium-sized enterprises. Nevertheless, this strategy offers a much higher ceiling in terms of efficiency and innovation potential. Once a robust chassis strain is established, it can be rapidly reprogrammed or modularly adjusted to produce a variety of bioactive peptides with different functions, significantly reducing time and cost for each new target. Moreover, the integration of machine learning and AI into strain design and peptide prediction workflows holds the promise of accelerating discovery, optimizing peptide properties, and expanding the functional diversity of the final products. In this era of digital biology, AI-powered peptide biomanufacturing systems could unlock unprecedented levels of precision, scalability, and customization in LAB-driven production platforms.

## 6. Conclusions

Lactic acid bacteria (LAB) represent a natural and versatile microbial platform for the targeted release of bioactive peptides from bovine β-casein, offering an effective strategy to valorize dairy proteins into functional ingredients. The sophisticated proteolytic system of LAB—particularly in strains like *Lactococcus lactis* and *Lactobacillus helveticus*—enables precise cleavage of casein sequences to liberate peptides with documented antihypertensive, antioxidative, antimicrobial, immunomodulatory, and metabolic regulatory functions.

These peptides contribute not only to the sensory and nutritional value of fermented dairy products but also hold promise as therapeutic agents in the context of functional foods and nutraceuticals. Despite substantial progress in peptide identification and characterization, challenges remain in optimizing peptide yield, specificity, and stability during industrial production. Recent advances in molecular tools such as CRISPR-based genome modulation (CRISPRi/a), inducible expression systems (e.g., NICE, Zirex, XIES, and agmatine-regulated promoters), and cell-free platforms have opened new avenues to engineer LAB for enhanced peptide liberation. Furthermore, multi-omics approaches allow for a systems-level understanding of LAB proteolysis and metabolic networks, offering data-driven insights into strain selection, process tuning, and pathway optimization.

With the convergence of synthetic biology, fermentation engineering, and computational biology, LAB-based biomanufacturing is poised to enter a new era of precision and efficiency. Synthetic biology, by enabling the rational design of LAB chassis, allows for the precise tailoring of microbial systems that can be optimized for specific proteolytic tasks. LAB strains can now be engineered to possess enhanced, targeted proteases that selectively cleave casein peptides with higher specificity and yield. This engineering opens the door to a range of bioactive peptides with improved bioavailability and potency. Enzymatic engineering is crucial in optimizing the processes by which these peptides are liberated, focusing on modulating the activity of endogenous proteases or introducing novel ones, thus enhancing peptide yield while preventing the production of unwanted byproducts such as bitter peptides. Furthermore, the integration of computational biology, coupled with AI-driven platforms, accelerates the design and optimization process. By using predictive models based on large datasets of fermentation variables, machine learning can help predict optimal fermentation conditions in real-time, improving scalability and consistency across production batches. These advances will allow for unprecedented control over peptide production, providing a clear path to scalability in industrial applications. As computational tools integrate multi-omics data, the feedback loop between real-time fermentation monitoring and process adjustments will enable continuous improvements to both strain design and fermentation parameters, leading to more efficient, cost-effective, and reproducible peptide production. Moreover, with the increasing ability to predict and tailor the bioactivity of peptides, these technologies hold the potential to support the development of highly specialized, personalized bioactive peptides for a variety of health-related applications, from chronic disease prevention to targeted gut microbiome modulation.

However, as we progress toward this new era, regulatory considerations remain a critical component in ensuring the safe and ethical deployment of GM-LAB in the commercial production of bioactive peptides. Regulatory frameworks across different regions present unique challenges, which often require extensive testing and detailed safety documentation, particularly for novel GM strains in food applications, including assessments of potential toxicity, genetic stability, and the absence of transferable antibiotic resistance markers. This creates a significant regulatory burden for companies seeking to commercialize GM-LAB-derived products, which can delay product introduction and increase costs. Furthermore, public perception of GMOs, especially in the food and health sectors, remains a significant challenge. Even with robust safety data, consumer skepticism can hinder market acceptance, particularly for sensitive product categories like infant nutrition. To mitigate these challenges, the convergence of synthetic biology, enzymatic engineering, computational biology, and regulatory policy will be essential. It is important to continue engaging with regulatory authorities to ensure clear, standardized guidelines for GM organisms in food production. Furthermore, public awareness campaigns and stakeholder engagement will play a vital role in building trust and acceptance of GM-LAB-derived products. Collaborations between industry stakeholders, regulatory bodies, and consumers will help streamline approval processes, reduce market entry barriers, and ensure that bioactive peptides are delivered safely, ethically, and efficiently to consumers.

As consumer interest grows in gut health, personalized nutrition, and sustainable protein use, LAB-derived bioactive peptides stand at the forefront of functional food innovation. Realizing their full potential will require continued interdisciplinary collaboration—bridging microbiology, food technology, biotechnology, and regulatory policy—to accelerate the transition from bench to product and, ultimately, deliver scientifically validated, safe, and effective peptide-based interventions to the global market.

## Figures and Tables

**Figure 2 ijms-26-08119-f002:**
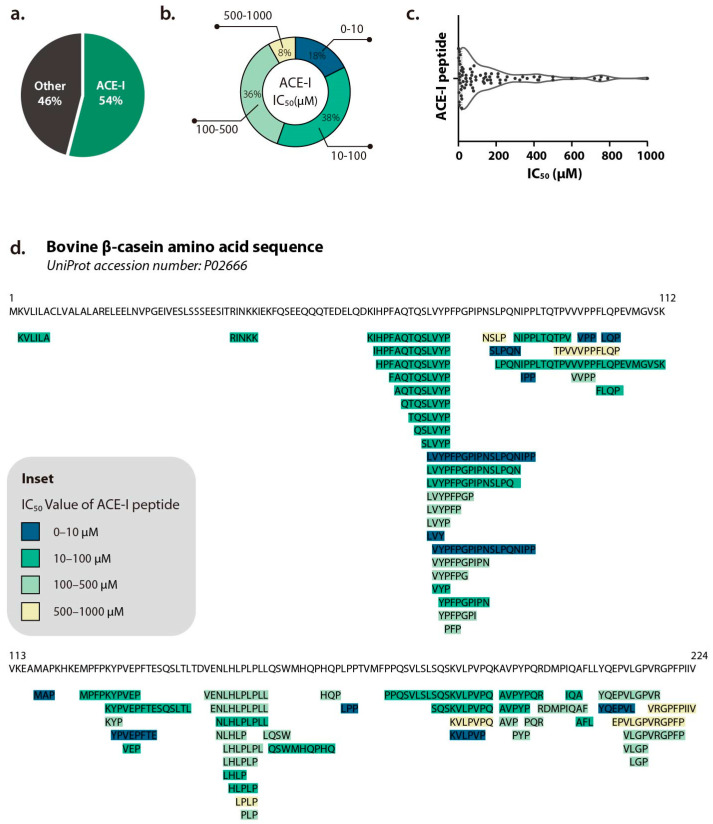
(**a**) Pie chart showing the proportion of Angiotensin-I-Converting Enzyme inhibitory (ACE-I) peptides among all bovine β-casein-derived bioactive peptides. (**b**) Doughnut chart visualizing the proportion of the half maximal inhibitory concentration (IC50) of all bovine β-casein-derived ACE-I peptides. (**c**) Violin plot of the IC50 value distribution of all bovine β-casein-derived ACE-I peptides. (**d**) Alignment all bovine β-casein-derived ACE-I peptides with bovine β-casein. Color of each peptide is based on its IC50 value, as indicated in the inset.

**Figure 3 ijms-26-08119-f003:**
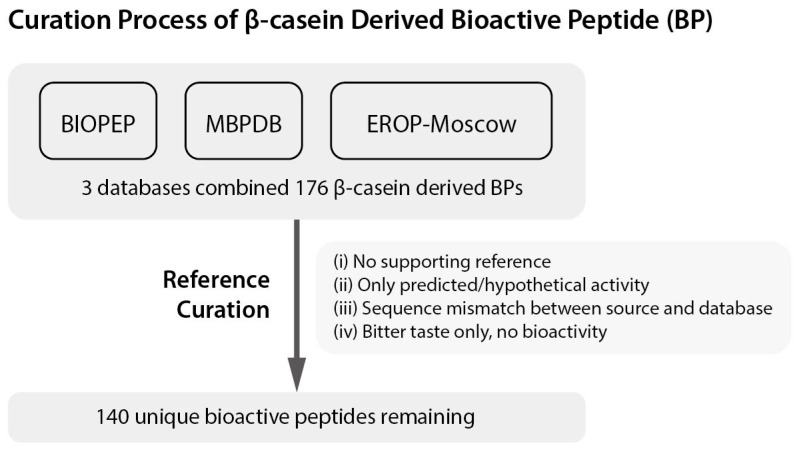
Flowchart of the curation process for β-casein-derived bioactive peptides. The diagram illustrates the screening process applied to peptides retrieved from three databases (BIOPEP, MBPDB, and EROP-Moscow). Four exclusion criteria were used: (i) lack of supporting references, (ii) only hypothetical or predicted activity, (iii) discrepancies between the peptide sequence and the original literature, and (iv) peptides associated solely with a bitter taste. The final selection resulted in 140 unique bioactive peptides, as listed in Table 1.

**Figure 4 ijms-26-08119-f004:**
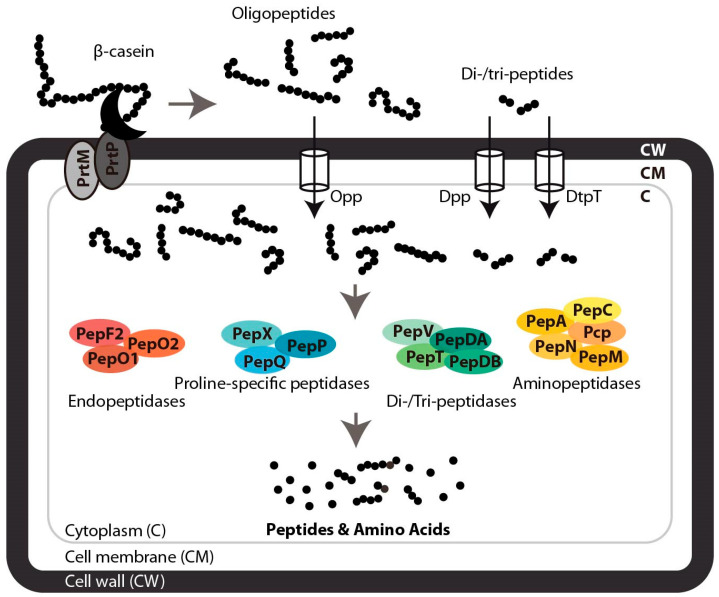
Schematic representation of the peptide liberation system of *Lc. lactis* MG1363 process from β-casein β-casein hydrolysis is initiated by PrtP, after its autoproteolytic activation with the aid of PrtM. Subsequently, oligopeptides are transported into the cell by the oligopeptide permease Opp, while di-/tripeptides are internalized by the Dpp or DtpT transport system [173]. The peptides are then degraded into smaller peptides and free amino acids by the concerted action of 15 peptidases, which are classified and colored by their indicated cleavage specificity [15].

**Figure 5 ijms-26-08119-f005:**
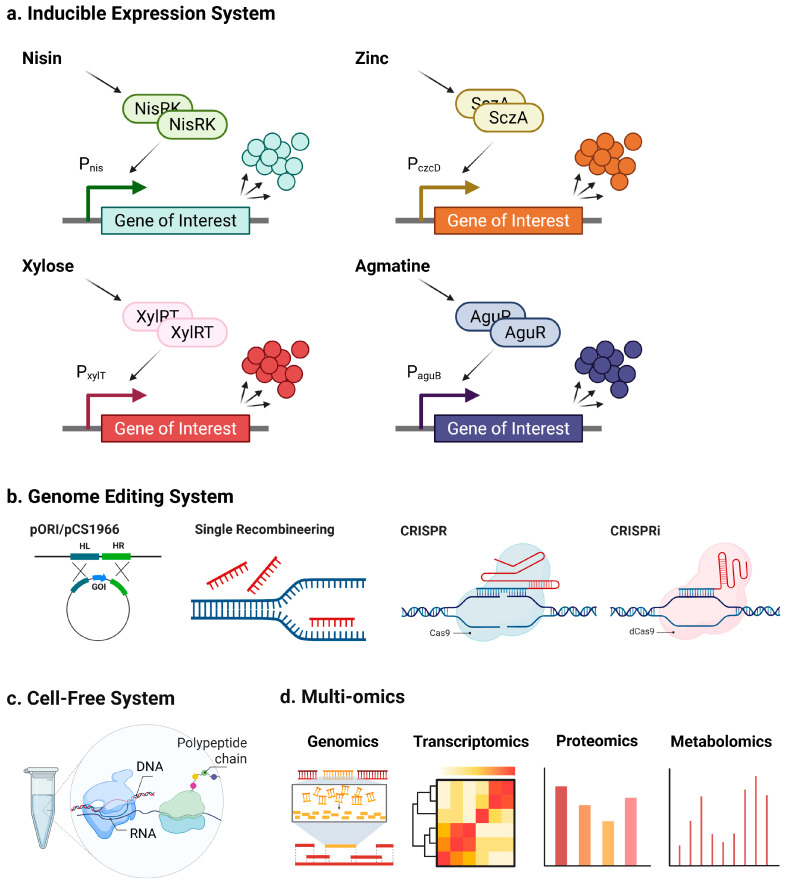
Engineered LAB-based strategies for enhanced peptide production: (**a**) Representative inducible expression systems commonly applied in LAB, such as nisin-controlled gene expression (NICE), xylose-, zinc-, and agmatine-inducible promoters. (**b**) Genome editing tools for LAB, including pORI- or pCS1966-based homologous recombination, RecT-mediated single-strand recombineering, and CRISPR(i)-based approaches. (**c**) Cell-free protein synthesis (CFPS) systems derived from LAB, enabling rapid peptide production and pathway prototyping without living cells. (**d**) Application of multi-omics approaches (genomics, transcriptomics, proteomics, and metabolomics) to uncover and optimize peptide biosynthesis pathways in LAB.

**Table 1 ijms-26-08119-t001:** Selection of 140 certified bioactive peptides derived from bovine β-casein.

NO	Sequence ^a^	Start ^b^	End ^b^	Bioactivity ^c^	References
1	KVLILA	2	7	ACE-I	[61]
2	RELEELNVPGEIVESLSSSEESITRINK	16	43	Immuno-R	[19,21,43,62,63,64,65,66,67]
3	RELEELNVPGEIVESLSSSEESITR	16	40	Immuno-R	[18,20,22,68]
4	LNVPGEIVE	21	29	ACE-I	[69]
5	VPGEIVE	23	29	DPP-IV-I	[70]
6	RINKK	40	44	ACE-I; Anti-M	[71,72]
7	RINK	40	43	Anti-M	[71]
8	INKKI	41	45	Immuno-R; Anti-C	[73,74]
9	NKKI	42	45	Anti-M	[71]
10	DELQDKIHPFAQTQSLVYPFPGPIPNS	58	84	ACE-I	[75]
11	DKIHPF	62	67	ACE-I	[69]
12	KIHPFAQTQSLVYP	63	76	ACE-I	[76]
13	IHPFAQTQ	64	71	PEP-I	[77]
14	IHPFAQTQSLVYP	64	76	ACE-I	[76]
15	HPFAQTQSLVYP	65	76	ACE-I	[76]
16	FAQTQSLVYP	67	76	ACE-I	[76]
17	AQTQSLVYP	68	76	ACE-I	[76]
18	QTQSLVYP	69	76	ACE-I	[76]
19	TQSLVYP	70	76	ACE-I	[76]
20	QSLVYP	71	76	ACE-I	[76]
21	SLVYP	72	76	ACE-I	[76]
22	LVYPFPGPIPNSLPQNIPP	73	91	ACE-I	[78,79,80]
23	LVYPFPGPIPNSLPQN	73	88	ACE-I	[81]
24	LVYPFPGPIPNSLPQ	73	87	PEP-I	[53]
25	LVYPFPGP	73	80	ACE-I	[78]
26	LVYPFP	73	78	ACE-I	[82]
27	LVYP	73	76	ACE-I	[76]
28	LVY	73	75	ACE-I	[83]
29	VYPFPGPIP	74	82	PEP-I	[77]
30	VYPFPGPI	74	81	PEP-I	[77]
31	VYPFPGPIPN	74	83	ACE-I	[84]
32	VYPFPGPIPNSLPQNIPP	74	91	ACE-I	[79]
33	VYPFPG	74	79	ACE-I	[85]
34	VYP	74	76	ACE-I	[76,85]
35	YPFPGPIP	75	82	Opioid; ACE-I	[86,87]
36	YPFPGPIPNSL	75	85	Opioid	[88,89]
37	YPFPGPI	75	81	Opioid; Satiety-Ic; Immuno-R; Anxiolytic; Anti-C	[25,26,27,28,90,91,92,93,94,95,96,97,98,99]
38	YPFPG	75	79	Opioid; PNO; IM-Ic; Immuno-R; L&M-Ip	[25,31,33,34,91,92]
39	YPFPGP	75	80	Opioid; DPP-IV-I	[25,70,90,100]
40	YPFP	75	78	Opioid; Anti-C	[90,99,100,101]
41	YPFPGPIPN	75	83	ACE-I; DPP-IV-I; Opioid	[84,102,103,104]
42	YPFPGPIPNSLPQ	75	87	Opioid	[102]
43	YPFPGPIPNSLPQNIPPLTQT	75	95	Opioid	[102]
44	PFPGPI	76	81	Cathepsin B-I	[90]
45	FPGPIPN	77	83	DPP-IV-I	[104]
46	PGPIPN	78	83	Immuno-R; Anti-C	[105,106,107,108,109,110]
47	NSLP	83	86	ACE-I	[111]
48	SLPQN	84	88	ACE-I	[72]
49	LPQNIPPL	85	92	DPP-IV-I	[104]
50	LPQNIPPLT	85	93	DPP-IV-I	[70]
51	LPQNIPP	85	91	DPP-IV-I	[104]
52	LPQNIPPLTQTPVVVPPFLQPEVMGVSK	85	112	ACE-I	[75]
53	LPQ	85	87	DPP-IV-I	[104]
54	PQNIPPL	86	92	DPP-IV-I	[104]
55	NIPPLTQTPV	88	97	ACE-I	[69]
56	IPPLTQT	89	95	DPP-IV-I	[70]
57	IPP	89	91	ACE-I	[49]
58	LTQTPVVVPPF	92	102	ACE-I	[80,112]
59	TQTPVVVPPFLQPE	93	106	Anti-O (DPPH)	[52]
60	TPVVVPPFLQP	95	105	ACE-I	[85]
61	VVVPPF	97	102	ACE-I	[80]
62	VVPP	98	101	ACE-I	[61]
63	VPP	99	101	ACE-I; Anti-Infla; Boneloss-R	[49,72,113,114,115,116,117,118,119,120,121]
64	FLQP	102	105	ACE-I; DPP-IV-I	[111,122]
65	LQP	103	105	ACE-I	[51]
66	LQPE	103	106	Hypolipidemic	[55]
67	GVSKVKEAMAPKHKEMPFPKYPVEPFTESQ	109	138	Opioid; MUC-Ic	[123,124,125,126]
68	VKEAMAPK	113	120	Anti-O (LA/Lox; LA/AAPH; Hpode/Hb); Anti-M	[40,127]
69	EAMAPKHK	115	122	Anti-M	[40]
70	EAMAPK	115	120	Anti-M	[40]
71	MAP	117	119	ACE-I	[51,111]
72	HKEMPFPK	121	128	Anti-M	[40,128,129]
73	EMPFPK	123	128	MUC-Ic; Anti-M; Anti-M; Brad-P	[40,124,130,131]
74	MPFPKYPVEP	124	133	ACE-I	[132]
75	KYPVEPFTESQSLTL	128	142	ACE-I	[75]
76	KYP	128	130	ACE-I	[111]
77	YPVEPF	129	134	Opioid; DPP-IV-I; MUC-Ic	[70,102,124,133]
78	YPVEPFTE	129	136	ACE-I; Brad-P	[131]
79	VEP	131	133	ACE-I	[111]
80	HLPLP	140	144	ACE-I	[76,134,135]
81	LPLP	143	146	ACE-I	[76]
82	VENLHLPLPLL	145	155	ACE-I	[136]
83	ENLHLPLPLL	146	155	ACE-I	[136]
84	NLHLP	147	151	ACE-I	[76]
85	NLHLPLPLL	147	155	ACE-I	[136]
86	LHLP	148	151	ACE-I	[76]
87	LHLPLPL	148	154	ACE-I	[80]
88	LHLPLP	148	153	ACE-I	[80]
89	LPLPLL	150	155	DPP-IV-I	[70]
90	LPLPL	150	154	DPP-IV-I	[70,137]
91	LPL	150	152	DPP-IV-I	[137]
92	LQSW	155	158	ACE-I	[138]
93	QSWMHQPHQ	156	164	ACE-I	[139]
94	HQP	163	165	ACE-I	[111]
95	PLP	165	167	ACE-I	[76]
96	LPP	166	168	ACE-I	[111]
97	PPQSVLSLSQSKVLPVPQ	173	190	ACE-I	[75]
98	SQSKVLPVPQ	181	190	ACE-I	[132]
99	SQSKVLPVPQK	181	191	Hypolipidemic	[57]
100	SKVLPVPQ	183	190	ACE-I	[75]
101	KVLPVPQK	184	191	Anti-O (LA/Lox; LA/AAPH; Hpode/Hb)	[127]
102	KVLPVPQ	184	190	ACE-I	[138]
103	KVLPVP	184	189	ACE-I	[138,140]
104	VLPVPQ	185	190	Hypolipidemic	[55]
105	VLPVPQK	185	191	Anti-O (LA/Lox; LA/AAPH; Hpode/Hb); Anti-M; Anti-apo	[40,127,141]
106	VLPVPQKAVPYPQR	185	198	Anti-M	[40]
107	LPVPQ	186	190	DPP-IV-I	[70]
108	LPVP	186	189	DPP-IV-I	[133]
109	AVPYPQR	192	198	ACE-I; Anti-O (LA/Lox; LA/AAPH; Hpode/Hb); Anti-M	[40,127,130,142]
110	AVPYP	192	196	ACE-I	[143]
111	AVP	192	194	ACE-I	[143]
112	VPYPQ	193	197	Anti-O (ORAC)	[144]
113	PYPQ	194	197	Anti-O (ABTS)	[145]
114	PYP	194	196	ACE-I	[143]
115	PQR	196	198	ACE-I	[146]
116	RDMPIQAF	198	205	ACE-I	[75]
117	DMPIQAFLLYQEPVLGPVR	199	217	Anti-Infla	[147]
118	IQA	202	204	ACE-I	[111]
119	AFL	204	206	ACE-I	[148]
120	LLYQEPVLGPVRGPFPIIV	206	224	ACE-I	[75]
121	LLY	206	208	Immuno-R	[149]
122	LYQEPVLGPVRGPFPIIV	207	224	Immuno-R	[150]
123	YQEPVLGPVRGPFPI	208	222	Anti-M	[39]
124	YQEPVLGPVRGPFPIIV	208	224	ACE-I; Anti-M; Immuno-R	[39,75,151,152]
125	YQEPVLGPVR	208	217	ACE-I; Anti-O (ABTS&ORAC); Anti-Infla; Anti-Co	[54,153,154]
126	YQEPVL	208	213	ACE-I	[72,130]
127	QEPVL	209	213	Anti-Infla; Immuno-R	[44,155]
128	QEPV	209	212	Immuno-R	[155]
129	QEPVLGPVRGPFP	209	221	Hypoglycemic	[56]
130	QEPVLGPVRGPFPIIV	209	224	ACE-I	[153]
131	EPVLGPVRGP	210	219	Cyto-M	[156]
132	EPVLGPVRGPFP	210	221	ACE-I	[132]
133	VLGP	212	215	ACE-I; DPP-IV-I	[111,122]
134	VLGPVRGPFP	212	221	ACE-I	[80]
135	LGP	213	215	ACE-I	[157,158]
136	VRGPFPIIV	216	224	ACE-I	[80]
137	VRGPFP	216	221	ACE-I	[159]
138	GPFPIIV	218	224	ACE-I	[160,161]
139	GPFPI	218	222	Cathepsin B-I	[162]
140	PFP	219	221	ACE-I	[163]

^a^ The peptide sequences are presented in the one-letter amino acid code. ^b^ The start and end positions of the peptide from bovine β-casein sequence, including signal peptide, sequence from UniProt (www.uniprot.org (accessed on 15 May 2025)) with accession number P02666. ^c^ Bioactivities of the indicated peptide. Abbreviations: ACE-I, angiotensin-converting enzyme inhibitory; Anti-M, antimicrobial; Anti-O, antioxida-tive; Anti-apo, Anti-apoptotic; DPP-IV-I, dipeptidyl peptidase 4 inhibitory; Immuno-R, immunoregulatory; Anti-C, anticancer; PEP-I, prolyl endopeptidase-inhibitory; Satiety-Ic, Satiety Increasement; IM-Ic, Intestinal Mobility Increasement; PNO, Promoting Neurite Outgrowth; L&M-Ip, Learning and Memory Improvement; Cathepsin B-I, Cathepsin B inhibitory; Inflam-P, Inflammation Promotion; Boneloss-R, Bone loss reduction; Anti-Infla, Anti-inflammatory; MUC-Ic, MUC expression increasement; Brad-P, Bradykinin-Potentiating; Anti-Co, Anticoagulant; Cyto-M, Cytomodulatory.

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
