# Peer review of "Unlocking Casein Bioactivity: Lactic Acid Bacteria and Molecular Strategies for Peptide Release"

_ijms, 2025, doi:10.3390/ijms26178119_

Round 1

Reviewer 1 Report

Comments and Suggestions for Authors

The manuscript addresses an important topic with strong potential, but revisions to improve clarity, depth, and logical flow are needed. I recommend reconsideration after incorporating these suggestions.

  1. The current flow between Section 3 ("Proteolytic Systems of Lactic Acid Bacteria") and Section 4 ("Molecular Strategies") lacks a clear bridge. It is recommended to add a concluding paragraph at the end of Section 3 that explicitly summarizes the limitations of wild-type LAB (e.g., strain-dependent peptide yields, inefficient targeting of specific bioactive sequences, and susceptibility to over-hydrolysis of target peptides). This will better contextualize the necessity of the molecular engineering strategies introduced in Section 4, enhancing overall logical continuity.

  1. Figure 1C contains the phrase "benefiting men," which is imprecise. Could it be revised to "benefiting human health."

  1. Table 1 lists 140 peptides but lacks detailed descriptions of the screening criteria in the main text.

  1. Section 2.6 mentions that MAP has an IC50 of 0.8 μM but does not specify the experimental context (e.g., in vitro ACE inhibition assay vs. in vivo models). Please clarify this and include a brief comparison with other well-characterized ACE-inhibitory peptides (e.g., VPP and IPP) to highlight its potency.

  1. Section 4.4 mentions "multi-omics to unravel peptide biosynthesis" but lacks concrete examples. Please add 1–2 case studies (e.g., a transcriptomic analysis identifying key protease genes in mixed LAB fermentation or metabolomic insights into peptide accumulation pathways) to illustrate practical applications.

  1. In the future perspectives, elaborate on interdisciplinary collaboration (e.g., how food engineers and computational biologists can jointly optimize fermentation parameters for peptide production).

Author Response

The manuscript addresses an important topic with strong potential, but revisions to improve clarity, depth, and logical flow are needed. I recommend reconsideration after incorporating these suggestions.

Comment 1: The current flow between Section 3 ("Proteolytic Systems of Lactic Acid Bacteria") and Section 4 ("Molecular Strategies") lacks a clear bridge. It is recommended to add a concluding paragraph at the end of Section 3 that explicitly summarizes the limitations of wild-type LAB (e.g., strain-dependent peptide yields, inefficient targeting of specific bioactive sequences, and susceptibility to over-hydrolysis of target peptides). This will better contextualize the necessity of the molecular engineering strategies introduced in Section 4, enhancing overall logical continuity.

Response 1: We really appreciate the elaborate review and valuable comments raised by the reviewer. As suggested by the reviewer, we have added a new concluding paragraph at the end of Section 3. The revised details are as follows: “However, when redirected toward the intentional production of specific bioactive peptides, native LAB systems reveal notable limitations. These include strain-dependent variability in protease activity, low or inconsistent yields of target peptides, and limited specificity for cleaving desired sequences. In addition, uncontrolled proteolysis can result in over-hydrolysis, degrading functional peptides and reducing their bioactivity. This unpredictability hampers both mechanistic studies and industrial applications, where reproducibility, efficacy, and scalability are essential. These challenges underscore the need for rationally designed LAB strains with greater control over proteolytic precision. In the following section, we highlight emerging molecular strategies—such as genome editing, synthetic biology, and omics-driven optimization—designed to overcome these limitations and fully harness LAB’s biosynthetic potential for targeted bioactive peptide production.” This statement has been added to the revised manuscript at lines 312-323 (highlighted in red).

Comment 2: Figure 1C contains the phrase "benefiting men," which is imprecise. Could it be revised to "benefiting human health."

Response 2: Thank you for pointing this out. We fully agree that the original phrasing could lead to misinterpretation and does not reflect inclusive language. We have accordingly revised the phrase in Figure 1C from “benefiting men” to “benefiting human health,” ensuring both clarity and inclusivity in our representation of the figure’s intent. See the revised manuscript at line 95(highlighted in red).

Comment 3: Table 1 lists 140 peptides but lacks detailed descriptions of the screening criteria in the main text.

Response 3: Thanks for the reviewer’s helpful suggestion. Based on reviewer’s recommendation, we have expanded the relevant section in the manuscript as follows: “To ensure data quality and relevance, we applied four exclusion criteria: (i) no supporting reference; (ii) only hypothetical or predicted activity; (iii) discrepancies between database sequences and original literature; and (iv) bitter taste reported without any functional bioactivity” (see lines 214-217 in the revised manuscript). Additionally, we have incorporated a new schematic diagram (Figure R1 or Figure 3; see lines 223229) that outlines the overall selection framework, including the original data sources and the quality control measures applied. We believe this addition will help readers better understand the methodological rigor and transparency underlying Table 1.

Figure R1. Flowchart of the curation process for β-casein-derived bioactive peptides. The diagram illustrates the screening process applied to peptides retrieved from three databases (BIOPEP, MBPDB, and EROP-Moscow). Four exclusion criteria were used: (i) lack of supporting references, (ii) only hypothetical or predicted activity, (iii) discrepancies between the peptide sequence and the original literature, and (iv) peptides associated solely with a bitter taste. The final selection resulted in 140 unique bioactive peptides, as listed in Table 1.

Comment 4: Section 2.6 mentions that MAP has an IC50 of 0.8 μM but does not specify the experimental context (e.g., in vitro ACE inhibition assay vs. in vivo models). Please clarify this and include a brief comparison with other well-characterized ACE-inhibitory peptides (e.g., VPP and IPP) to highlight its potency.

Response 4: We appreciate the reviewer’s valuable suggestion. As the reviewer suggested, we have revised Section 2.6 to clarify the experimental context of the ACE inhibitory activity of Met-Ala-Pro (MAP) on Page 5: “The IC50 values of VPP and IPP are reported to be 9 μM and 5 μM, respectively, in an in vitro ACE inhibition assay. Identified as the most potent peptide to date, Met-Ala-Pro (MAP) exhibits an ICâ‚…â‚€ of 0.8 μM in an in vitro ACE inhibition assay and has also demonstrated pronounced blood pressure-lowering effects in an in vivo hypertensive rat model (3 mg/kg dose, P<0.01 at 8 hours post-administration)”. These revisions have been incorporated in the manuscript at lines 183-187 (highlighted in red).

Comment 5: Section 4.4 mentions "multi-omics to unravel peptide biosynthesis" but lacks concrete examples. Please add 1–2 case studies (e.g., a transcriptomic analysis identifying key protease genes in mixed LAB fermentation or metabolomic insights into peptide accumulation pathways) to illustrate practical applications.

Response 5: We appreciate the reviewer’s valuable suggestion to include concrete examples to illustrate the practical applications of multi-omics in peptide biosynthesis. In response, we have expanded Section 4.4 to include two relevant case studies that demonstrate the utility of metabolomics and peptidomics in unraveling peptide biosynthesis during fermentation processes at lines 454-474 and highlighted in red: “For instance, metabolomic and peptidomic analyses have been used to identify bioactive peptides released during fermentation with novel probiotics like Lacticaseibacillus rhamnosus and Lactiplantibacillus plantarum, which revealed significant upregulation of ACE-I peptides. In a study of fresh cheese fermented with three novel probiotics, 112 bioactive peptides were significantly up-regulated, highlighting the potential of metabolomics and peptidomics for understanding peptide biosynthesis [201]. Additionally, an integrated approach using metabolomics and peptidomics was employed to delineate the characteristic metabolites and peptides in milk fermented with Lpb. plantarum L3. This study revealed that Lpb. plantarum L3 secreted a complex array of peptidases that cleaved bovine caseins at specific sites to release bioactive peptides, including ACE-I peptides, antioxidant peptides, and antimicrobial peptides. The enzyme systems secreted by Lpb. plantarum L3 preferentially cleaved lysine-based peptide bonds, and the released peptides exhibited various bioactivities [202].

These findings underscore the importance of multi-omics approaches in optimizing fermentation conditions to enhance the functional properties of fermented dairy products. By combining the metabolomic and peptidomic data, this integrated multi-omics approach provides valuable insights into the complex enzymatic processes that regulate the release of bioactive peptides during fermentation. Such knowledge can be applied to optimize LAB strains for specific peptide production, leading to improved functional foods with targeted health benefits.

Comment 6: In the future perspectives, elaborate on interdisciplinary collaboration (e.g., how food engineers and computational biologists can jointly optimize fermentation parameters for peptide production).

Response 6: Thank you for your valuable feedback. We appreciate the suggestion to elaborate on the interdisciplinary collaboration between food engineers and computational biologists. In response, we have expanded the discussion in the "Future Perspectives" section to emphasize the importance of such collaborations at lines 526-562 and highlighted in red: “Looking forward, the convergence of synthetic biology, artificial intelligence (AI), and precision fermentation is expected to define the next frontier in LAB-based bioactive peptide development. Synthetic biology offers the possibility of designing “next-generation” LAB chassis tailored for controlled proteolysis and defined peptide profiles. To fully realize this potential, interdisciplinary collaboration between food engineers, computational biologists, and fermentation experts will be crucial. Food engineers can bring expertise in optimizing fermentation conditions, scaling up production, and ensuring the consistency of peptide output, while computational biologists will focus on creating predictive models, integrating omics data, and using AI tools to optimize peptide sequence design and bioactivity predictions. This collaboration will ensure that both the fermentation processes and the bioinformatics approaches are aligned to maximize the efficiency and specificity of peptide production. Coupled with multi-omics and high-throughput screening platforms, rational design of LAB protease repertoires could achieve both diversity and specificity in peptide output. Moreover, food engineers will play a pivotal role in translating computational predictions into practical applications by optimizing fermentation conditions such as nutrient availability, temperature, pH, and oxygen levels, which are crucial for the desired expression of bioactive peptides. Meanwhile, computational biologists, aided by machine learning algorithms, can continually improve their models by analyzing the resulting data from fermentation trials. This dynamic feedback loop will allow for a more iterative and fine-tuned approach to peptide production. AI-driven peptide discovery tools—trained on large datasets of sequence–function relationships—are being increasingly applied to predict peptide bioactivities and optimize sequences with enhanced stability, bioavailability, or receptor binding affinity. The synergy between computational biology and fermentation science will allow for the development of peptides with more predictable and controllable characteristics, ultimately reducing the time and resources required for optimization.

Furthermore, precision fermentation platforms promise reproducible, scalable, and cost-effective peptide production with minimal batch variability. To maximize the impact of these platforms, both computational biologists and food engineers will need to work together to fine-tune the optimization parameters and integrate real-time monitoring technologies, ensuring that the fermentation process remains robust and adaptable to various production scales. In the era of personalized nutrition and microbiome-aware interventions, collaborative efforts between these interdisciplinary teams will also facilitate the development of modular LAB consortia that can be designed to generate peptide profiles specifically tailored to individual health needs or microbiota compositions. This flexibility will allow for personalized approaches to peptide delivery, enhancing the efficacy of bioactive peptides in promoting human health and well-being”.

Reviewer 2 Report

Comments and Suggestions for Authors

The review is well-structured, segmented by important and relevant topics. The authors cover topics ranging from the production and consumption of milk from different species to the analysis of milk composition, particularly its protein content. They then addressed the bioactive potential of the various proteins, considering the sequence of the 140 active peptides derived from beta-casein.

The subsequent review, including the mechanism of action of LABs, was comprehensive and explanatory, providing the reader with an overview of their application in the delivery of active peptides, covering several areas, and finally offering future perspectives. Thus, the article provides a significant quantity and quality of references that supported a robust and consistent review.

Author Response

Comment 1: The review is well-structured, segmented by important and relevant topics. The authors cover topics ranging from the production and consumption of milk from different species to the analysis of milk composition, particularly its protein content. They then addressed the bioactive potential of the various proteins, considering the sequence of the 140 active peptides derived from beta-casein. 

The subsequent review, including the mechanism of action of LABs, was comprehensive and explanatory, providing the reader with an overview of their application in the delivery of active peptides, covering several areas, and finally offering future perspectives. Thus, the article provides a significant quantity and quality of references that supported a robust and consistent review. 

Response 1: We sincerely thank the reviewer for the thoughtful and positive feedback. We are grateful for your appreciation of the structure, content, and references within the review. Your comments validate our efforts to present a comprehensive and coherent overview of the various topics related to β-casein-derived bioactive peptides and the role of LABs in their delivery. We are glad that the manuscript met your expectations in terms of clarity, depth, and the integration of key references. We will continue to refine the manuscript and ensure it maintains the highest standard of quality. Thank you again for your constructive and encouraging review.

Reviewer 3 Report

Comments and Suggestions for Authors

The presented material seems interesting and requires partial revision.
The authors need to make minor adjustments to improve the article.
The comments are provided below.
1. Provide a generalized scientific goal of the research in the abstract of the article
2. It is necessary to provide a comprehensive diagram of the results of the theoretical study of the material
3. When describing the antioxidant effects (422), antioxidant activity (282), it is necessary to add material indicating the methods for determining these properties, since these indicators vary significantly and differ from the methods and are described in a number of cited publications (sources 52, 145, 128, etc.)
4. From my point of view, point 5. "Problems, regulatory aspects and future prospects" is not sufficiently disclosed; it is necessary to cover these issues in more detail, since this characterizes the prospects of the described direction of research
5. It is also necessary to more fully disclose the idea in the conclusions regarding the fact that the convergence of synthetic biology, enzymatic engineering and computational biology using LAB is ready to enter a new era of accuracy and efficiency.
6. The article must provide data on the financial costs of implementing these developments in industry, how expensive this approach is

Author Response

The presented material seems interesting and requires partial revision. The authors need to make minor adjustments to improve the article. The comments are provided below.

Comment 1: Provide a generalized scientific goal of the research in the abstract of the article.

Response 1: We thank the reviewer for this helpful suggestion. In response, we have revised the abstract to include a more explicit statement of the generalized scientific goal of the study at lines 16–19 in red: “the scientific goal of this study is to provide a comprehensive synthesis of how synthetic biology, molecular biotechnology, and systems-level approaches can be leveraged to enhance the targeted discovery and production of β-casein-derived bioactive peptides”. 

Comment 2: It is necessary to provide a comprehensive diagram of the results of the theoretical study of the material

Response 2: We appreciate the reviewer’s valuable recommendation to include a visual summary of our theoretical analysis. Accordingly, we have incorporated a new schematic diagram (Figure R1 or Figure 3; see lines 223-229) that outlines the overall selection framework, including the original data sources and the quality control measures applied. We believe this addition enhances the transparency and accessibility of the methodology.

Figure R1. Flowchart of the curation process for β-casein-derived bioactive peptides. The diagram illustrates the screening process applied to peptides retrieved from three databases (BIOPEP, MBPDB, and EROP-Moscow). Four exclusion criteria were used: (i) lack of supporting references, (ii) only hypothetical or predicted activity, (iii) discrepancies between the peptide sequence and the original literature, and (iv) peptides associated solely with a bitter taste. The final selection resulted in 140 unique bioactive peptides, as listed in Table 1.

Comment 3: When describing the antioxidant effects (422), antioxidant activity (282), it is necessary to add material indicating the methods for determining these properties, since these indicators vary significantly and differ from the methods and are described in a number of cited publications (sources 52, 145, 128, etc.)

Response 3: Thank you for this insightful comment. We fully agree that antioxidant activity can be evaluated using a variety of assays, and that the results are highly dependent on the specific methodologies employed. To address this, we have revised Table 1 to explicitly indicate the antioxidant assessment methods used in each referenced study—for example, DPPH, ABTS, ORAC, and lipid peroxidation inhibition assays such as LA/LOX, AAPH, and Hb/Hpode. This clarification aims to help readers better understand the experimental context and enhance the interpretability and comparability of antioxidant findings across the literature.

The “Anti-O” activity in Table 1 (marked in red at line 230) has been updated accordingly, as follows:

NO

Sequence a

Start b

End b

Bioactivity c

Reference

59

TQTPVVVPPFLQPE

93

106

Anti-O (DPPH)

[52]

68

VKEAMAPK

113

120

Anti-O (LA/Lox; LA/AAPH; Hpode/Hb); Anti-M

[40,127]

101

KVLPVPQK

184

191

Anti-O (LA/Lox; LA/AAPH; Hpode/Hb)

[127]

105

VLPVPQK

185

191

Anti-O (LA/Lox; LA/AAPH; Hpode/Hb); Anti-M; Anti-apo

[40,127,141]

109

AVPYPQR

192

198

ACE-I; Anti-O (LA/Lox; LA/AAPH; Hpode/Hb); Anti-M

[40,127,130,142]

112

VPYPQ

193

197

Anti-O (ORAC)

[144]

113

PYPQ

194

197

Anti-O (ABTS)

[145]

125

YQEPVLGPVR

208

217

ACE-I; Anti-O (ABTS&ORAC); Anti-Infla; Anti-Co

[54,153,154]

Comment 4: From my point of view, point 5. "Problems, regulatory aspects and future prospects" is not sufficiently disclosed; it is necessary to cover these issues in more detail, since this characterizes the prospects of the described direction of research

Response 4: Thank you for your thoughtful feedback. In response to your comment, we have significantly expanded the "Challenges, Regulatory Considerations, and Future Perspectives" section in lines 526-562 in red: “Looking forward, the convergence of synthetic biology, artificial intelligence (AI), and precision fermentation is expected to define the next frontier in LAB-based bioactive peptide development. Synthetic biology offers the possibility of designing “next-generation” LAB chassis tailored for controlled proteolysis and defined peptide profiles. To fully realize this potential, interdisciplinary collaboration between food engineers, computational biologists, and fermentation experts will be crucial. Food engineers can bring expertise in optimizing fermentation conditions, scaling up production, and ensuring the consistency of peptide output, while computational biologists will focus on creating predictive models, integrating omics data, and using AI tools to optimize peptide sequence design and bioactivity predictions. This collaboration will ensure that both the fermentation processes and the bioinformatics approaches are aligned to maximize the efficiency and specificity of peptide production. Coupled with multi-omics and high-throughput screening platforms, rational design of LAB protease repertoires could achieve both diversity and specificity in peptide output. Moreover, food engineers will play a pivotal role in translating computational predictions into practical applications by optimizing fermentation conditions such as nutrient availability, temperature, pH, and oxygen levels, which are crucial for the desired expression of bioactive peptides. Meanwhile, computational biologists, aided by machine learning algorithms, can continually improve their models by analyzing the resulting data from fermentation trials. This dynamic feedback loop will allow for a more iterative and fine-tuned approach to peptide production. AI-driven peptide discovery tools—trained on large datasets of sequence–function relationships—are being increasingly applied to predict peptide bioactivities and optimize sequences with enhanced stability, bioavailability, or receptor binding affinity. The synergy between computational biology and fermentation science will allow for the development of peptides with more predictable and controllable characteristics, ultimately reducing the time and resources required for optimization.

Furthermore, precision fermentation platforms promise reproducible, scalable, and cost-effective peptide production with minimal batch variability. To maximize the impact of these platforms, both computational biologists and food engineers will need to work together to fine-tune the optimization parameters and integrate real-time monitoring technologies, ensuring that the fermentation process remains robust and adaptable to various production scales. In the era of personalized nutrition and microbi-ome-aware interventions, collaborative efforts between these interdisciplinary teams will also facilitate the development of modular LAB consortia that can be designed to generate peptide profiles specifically tailored to individual health needs or microbiota compositions. This flexibility will allow for personalized approaches to peptide delivery, enhancing the efficacy of bioactive peptides in promoting human health and well-being”. 

Comment 5: It is also necessary to more fully disclose the idea in the conclusions regarding the fact that the convergence of synthetic biology, enzymatic engineering and computational biology using LAB is ready to enter a new era of accuracy and efficiency.

Response 5: Thank you for your valuable comment. We have significantly expanded the conclusion section in lines 600-640 in red: “Synthetic biology, by enabling the rational design of LAB chassis, allows for the precise tailoring of microbial systems that can be optimized for specific proteolytic tasks. LAB strains can now be engineered to possess enhanced, targeted proteases that selectively cleave casein peptides with higher specificity and yield. This engineering opens the door to a range of bioactive peptides with improved bioavailability and potency. Enzymatic engineering is crucial in optimizing the processes by which these peptides are liberated, focusing on modulating the activity of endogenous proteases or introducing novel ones, thus enhancing peptide yield while preventing the production of unwanted byproducts such as bitter peptides. Furthermore, the integration of computational biology, coupled with AI-driven platforms, accelerates the design and optimization process. By using predictive models based on large datasets of fermentation variables, machine learning can help predict optimal fermentation conditions in real-time, improving scalability and consistency across production batches. These advances will allow for unprecedented control over peptide production, providing a clear path to scalability in industrial applications. As computational tools integrate multi-omics data, the feedback loop between real-time fermentation monitoring and process adjustments will enable continuous improvements to both strain design and fermentation parameters, leading to more efficient, cost-effective, and reproducible peptide production. Moreover, with the increasing ability to predict and tailor the bioactivity of peptides, these technologies hold the potential to support the development of highly specialized, personalized bioactive peptides for a variety of health-related applications, from chronic disease prevention to targeted gut microbiome modulation.

However, as we progress toward this new era, regulatory considerations remain a critical component in ensuring the safe and ethical deployment of GM-LAB in the commercial production of bioactive peptides. Regulatory frameworks across different regions present unique challenges, which often require extensive testing and detailed safety documentation, particularly for novel GM strains in food applications, including assessments of potential toxicity, genetic stability, and the absence of transferable antibiotic resistance markers. This creates a significant regulatory burden for companies seeking to commercialize GM-LAB-derived products, which can delay product introduction and increase costs. Furthermore, public perception of GMOs, especially in food and health sectors, remains a significant challenge. Even with robust safety data, consumer skepticism can hinder market acceptance, particularly for sensitive product categories like infant nutrition. To mitigate these challenges, the convergence of synthetic biology, enzymatic engineering, computational biology, and regulatory policy will be essential. It will be important to continue engaging with regulatory authorities to ensure clear, standardized guidelines for GM organisms in food production. Furthermore, public awareness campaigns and stakeholder engagement will play a vital role in building trust and acceptance of GM-LAB-derived products. Collaborations between industry stakeholders, regulatory bodies, and consumers will help streamline approval processes, reduce market entry barriers, and ensure that bioactive peptides are delivered safely, ethically, and efficiently to consumers.” 

Comment 6: The article must provide data on the financial costs of implementing these developments in industry, how expensive this approach is

Response 6: Thank you very much for your thoughtful comment regarding the financial costs associated with implementing LAB-based synthetic biology approaches in industry.

We fully agree that economic feasibility is a critical component of any translational biotechnology effort. In the revised manuscript, we have now added a paragraph in Section 5 to broadly discuss the cost landscape associated with these approaches. Specifically, we note that engineered strain development and regulatory compliance are currently the two main cost drivers. Although these initial investments may exceed those of conventional culture-dependent screening methods, they offer long-term advantages in efficiency, customizability, and scalability. To illustrate this point, we reviewed publicly available cost estimates. While exact figures are often proprietary and thus not easily disclosed in academic sources, we found a rough breakdown provided by FoodWrite (https://foodwrite.co.uk/costing-for-precision-fermentation/), which estimates that strain development can cost between $50,000 to $500,000+, scale-up for precision fermentation can cost between $100,000 to $1 million per target product. Regulatory approval can add another $500,000 to $2 million+ depending on the region and complexity of the submission.

Given the variability and commercial sensitivity of cost data, we decided not to include specific numbers in the main manuscript text. Instead, we provided a general discussion of where costs are concentrated and emphasized the longer-term benefits of reusability, AI-driven optimization, and the ability to iteratively produce novel peptides once a chassis LAB strain is established, in the manuscript as follows: “However, despite its scientific advantages, the current implementation of synthetic biology–assisted peptide production using LAB may involve higher upfront costs compared to conventional strain isolation and screening approaches. These expenses primarily arise from two domains: the development of engineered strains, including genetic modification and fermentation process optimization, and the subsequent investment in regulatory compliance, which is essential for product approval and market access. These stages require significant resource input and specialized expertise, posing a barrier to small- and medium-sized enterprises. Nevertheless, this strategy offers a much higher ceiling in terms of efficiency and innovation potential. Once a robust chassis strain is established, it can be rapidly reprogrammed or modularly adjusted to produce a variety of bioactive peptides with different functions, significantly reducing time and cost for each new target. Moreover, the integration of machine learning and AI into strain design and peptide prediction workflows holds the promise of accelerating discovery, optimizing peptide properties, and expanding the functional diversity of the final products. In this era of digital biology, AI-powered peptide biomanufacturing systems could unlock unprecedented levels of precision, scalability, and customization in LAB-driven production platforms”. The main text have been revised accordingly at lines 563-579 and highlighted in red.
